# Phonon Structure, Infra-Red and Raman Spectra of Li_2_MnO_3_ by First-Principles Calculations

**DOI:** 10.3390/ma15186237

**Published:** 2022-09-08

**Authors:** Ruth Pulido, Nelson Naveas, Raúl J. Martin-Palma, Fernando Agulló-Rueda, Victor R. Ferró, Jacobo Hernández-Montelongo, Gonzalo Recio-Sánchez, Ivan Brito, Miguel Manso-Silván

**Affiliations:** 1Departamento de Física Aplicada and Instituto Universitario de Ciencia de Materiales Nicolás Cabrera, Universidad Autónoma de Madrid, 28049 Madrid, Spain; 2Departamento de Ingeniería Química y Procesos de Minerales, Universidad de Antofagasta, Avenida Angamos 601, Antofagasta 1240000, Chile; 3Instituto de Ciencia de Materiales de Madrid (ICMM), CSIC, 28049 Madrid, Spain; 4Departamento de Ingeniería Química, Universidad Autónoma de Madrid, 28049 Madrid, Spain; 5Departamento de Ciencias Matemáticas y Físicas, UC Temuco, Temuco 4813302, Chile; 6Facultad de Ingeniería, Arquitectura y Diseño, Universidad de San Sebastián, Concepción 4080871, Chile; 7Departamento de Química, Universidad de Antofagasta, Avenida Angamos 601, Antofagasta 1240000, Chile; 8Centro de Microanálisis de Materiales, Universidad Autónoma de Madrid, Campus de Cantoblanco, 28049 Madrid, Spain

**Keywords:** Li_2_MnO_3_, DFT, DFPT, Raman spectroscopy, IR spectroscopy

## Abstract

The layer-structured monoclinic Li_2_MnO_3_ is a key material, mainly due to its role in Li-ion batteries and as a precursor for adsorbent used in lithium recovery from aqueous solutions. In the present work, we used first-principles calculations based on density functional theory (DFT) to study the crystal structure, optical phonon frequencies, infra-red (IR), and Raman active modes and compared the results with experimental data. First, Li_2_MnO_3_ powder was synthesized by the hydrothermal method and successively characterized by XRD, TEM, FTIR, and Raman spectroscopy. Secondly, by using Local Density Approximation (LDA), we carried out a DFT study of the crystal structure and electronic properties of Li_2_MnO_3_. Finally, we calculated the vibrational properties using Density Functional Perturbation Theory (DFPT). Our results show that simulated IR and Raman spectra agree well with the observed phonon structure. Additionally, the IR and Raman theoretical spectra show similar features compared to the experimental ones. This research is useful in investigations involving the physicochemical characterization of Li_2_MnO_3_ material.

## 1. Introduction

The growing demand for portable devices, electronic vehicles and energy storage, as well as attempts to mitigate the impacts of global warming, have led to rapid advances in the development of lithium-ion batteries as well as the search for new technologies in lithium recovery [1,2,3,4]. In this instance, lithium manganese oxide (LMO) materials have attracted a great deal of interest due to their key roles as cathodic and precursor materials for energy and lithium adsorption from aqueous solutions, respectively [5,6]. In the context of these potential applications, Li-rich LMO such as Li_2_MnO_3_ has been identified as a crucial component due to its interesting physicochemical properties [7,8]. Interestingly, it has been reported that the increase of Li_2_MnO_3_ content enhances the cycling stability of cathode materials based on Li-Mn-O, preventing the manganese dissolution [9,10]. Similarly, by controlling the Li_2_MnO_3_ layered phase content in lithium adsorbent precursor materials, it is possible to prevent the manganese dissolution from the LMO lattice during the Li^+^ desorption process [11].

The layer-structured monoclinic Li_2_MnO_3_ materials (with the rock salt crystal structure; Figure 1) belong to the C2/m space group. Li_2_MnO_3_ structure consists of laminations of alternating Li/Mn and Li-layers (Figure 1a). The Li^+^ and Mn^+4^ ions reside in the octahedral interstices of a close-packed cubic oxygen lattice stacking order, as represented in Figure 1b,c [12]. Their formula can also be written as:$${\left\{{\left(Li_{1/2}\right)}_{2c}{\left(Li_{1}\right)}_{4h}\right\}}_{interslab}\left\{{\left(Li_{1/2}\right)}_{2b}{\left(Mn_{1}\right)}_{4g}{\left(O_{1}\right)}_{4i}{\left(O_{2}\right)}_{8j}\right\}$$

The lithium ions are scattered octahedrally in the 2c and 4h Wyckoff position inside the [Li^+^] layer and in the 2b position into the [LiMn_2_] layer, while the Mn^+4^ ions are dispersed octahedrally in the 4g sites [13].

Li_2_MnO_3_-based LMO materials can be formed by different synthesis routes, varying in conformation and composition. Structures such as core-shell type, spinel embedded within the layered matrix, and spherical particles have been reported [9,14,15]. Whatever the conformation and composition are chosen, a detailed characterization of the LMO material is essential to provide an understanding of its different properties. For example, one of the most critical issues in understanding electrochemical properties is determining the local structure [16]. In this context, some studies have shown a relationship between the structural properties and electrochemical performance in doped-LMO composites [7,17]. Kim et al. reported that the enhanced stability of the local structure of lithium manganese oxides is related to the improvement of cyclability [18]. Likewise, the knowledge of the local environment of LMO helps to elucidate the underlying mechanism of the ad/desorption process in LMO-based lithium-ion sieves [19]. In this sense, vibrational spectroscopy can be used to provide information on the structural aspects of LMO. Raman and infra-red (IR) spectroscopies are very sensitive to the short-range environment of oxygen coordination around the cations [20]. Julien and co-workers have extensively studied the structural properties of spinel-type LMO by Raman and IR spectroscopies [20,21,22,23,24,25,26]. Their results are based on the experimental analysis of the LMO building blocks, namely, the tetrahedra LiO_4_ and the octahedra MnO_6_, which compose the crystal lattice. On the contrary, the vibrational analysis of the Li_2_MnO_3_ has not been completely and vigorously studied yet, despite the fact that its IR and/or Raman bands have been studied previously by experiments [8,27]. Therefore, there is a need to perform detailed studies in order to obtain solid knowledge about its vibrational properties. In this context, simulated IR and Raman spectra based on density functional theory (DFT) calculations can be helpful in understanding the vibrational properties of Li_2_MnO_3_. In fact, DFT-based theoretical studies have shown excellent agreement with experimental IR and/or Raman spectra of organic compounds [28], crystalline [29], and 2D materials [30].

In the present work, we report a computational study of the vibrational properties of Li_2_MnO_3_. Initially, we synthesized and physicochemically characterized the Li_2_MnO_3_ powder material. Further, we proceeded to the calculation of the optimized crystal structure and the electronic properties by first-principle calculations using the Local Density Approximation (LDA). Then, by using Density Functional Perturbation Theory (DFPT), we studied the vibrational modes and computed the phonon structure, IR and Raman spectra of the Li_2_MnO_3_. Finally, a detailed interpretation of obtained data was presented.

## 2. Materials and Methods

### 2.1. Material Synthesis and Characterization

The Li_2_MnO_3_ samples were synthesized by way of a hydrothermal method. First, LiOH (2 mol·L${}^{-1}$) and H_2_O_2_ (1.2 mol·L${}^{-1}$) were dissolved in ultra-pure water to produce Li precursor solutions. Then, MnSO_4_ (0.4 mol·L${}^{-1}$) was dissolved in ultra-pure water to form an Mn precursor solution. Both solutions were magnetically stirred for 2 h. Subsequently, the mixture was crystallized for 8 h at 180 ${}^{\circ }$C in a Teflon-lined stainless steel autoclave. The crystallization product was washed and centrifuged numerous times. Finally, the resultant product was dried at 60 ${}^{\circ }$C for 12 h before being calcined at 800 ${}^{\circ }$C for 4 h.

Powder X-ray Diffraction (XRD) in a Bruker New D8 Advance X-ray diffractometer equipped with a Johansson monochromator (Cu-Kα1 radiation (Billerica, MA, USA), λ = 1.5406 Å) and LYNXEYE XE detector was used to investigate the Li_2_MnO_3_ crystallographic structure. With an angular step of 0.02° and a scanning rate of 2${}^{\circ }$/s, the diffraction patterns were scanned from 2θ 10${}^{\circ }$ to 90${}^{\circ }$.

High-resolution transmission electron microscopy (HR-TEM) and electron diffraction (ED) (JEOL 2100F, Tokyo, Japan), operating under an accelerating voltage of 200 kV, were used to analyze the morphology and microstructure of Li_2_MnO_3_ samples.

The vibrational properties of Li_2_MnO_3_ samples were studied by Raman, and Fourier transform infrared (FT-IR) spectroscopies. Raman spectra were acquired from Renishaw Ramascope 2000 Raman microspectrometer, coupled to an Olympus BH-2 optical microscope at room temperature using argon ion laser (514.5 nm wavelength (green)) for excitation. The objective had a numerical aperture of NA = 0.80 and a magnification of 50×. On the sample surface, the laser power was of the order of 1 mW. Each CCD pixel integration time was 50 s. The FT-IR spectra were obtained from Spectrum Two Spectrometer (Perkin Elmer; Waltham, MA, USA) with a resolution of 4 cm^−1^ over the range of 4000–50 cm^−1^.

### 2.2. Calculation Details

The calculations were carried out under the Density Functional Theory (DFT) framework using the plane wave pseudopotential method coded inside the Quantum ESPRESSO computational package [31]. We used the conventional unit cell to represent Li_2_MnO_3_, which is composed of four formula units, resulting in a total of 24 atoms (Li_8_Mn_4_O_12_; see Figure 1b). Then, the optimized cell parameters were obtained via DFT by variable-cell relaxation. For Li, Mn, and O atoms, norm-conserving pseudopotentials based on Local Density Approximation (LDA) were selected [32]. We used very tight values for the kinetic energy cutoff, total energy and force tolerance, setting them to 80 Ry, 1 × 10${}^{-9}$ Ry and 1 × 10${}^{-5}$ Ry per Bohr, respectively. In order to sample the reciprocal space, we used the Monkhorst–Pack method [33] with a k-point mesh of 4 × 4 × 4 for the cell model, Li_8_Mn_4_O_12_. Despite the fact that the nature of Li_2_MnO_3_ is antiferromagnetic, several reports demonstrated that the effect of various spin configurations on the total formation energies is minimal; therefore, in order to simplify the model, only the ferromagnetic configuration is taken into account when the spin polarization is included in the calculations [13,34,35,36]. The optimized cell parameters and atomic positions were used to construct the electronic density of states (DOS) and the electronic band structure. In order to calculate the vibrational properties, we performed Density Functional Perturbation Theory (DFPT) using the Phonon Package of Quantum ESPRESSO [37]. Using the DFPT linear response approach at the Γ point, the dynamic matrix and frequencies of the harmonic phonon modes were determined. The calculated IR and Raman spectra were represented by a Gaussian function with an FWHM of 10 cm^−1^. The vibrational modes were visualized with the XCrysDen software [38]. The phonon band structure and phonon DOS were computed by using a q-point mesh of 2 × 2 × 2.

## 3. Results and Discussion

### 3.1. Characterization of the Li_2_MnO_3_ Powder Material

The performance of the LMO is strongly influenced by the crystalline structure, particle size, morphology and specific surface area. Therefore, it is important to conduct a detailed characterization before its application [11]. In order to characterize synthesized Li_2_MnO_3_ in terms of crystalline structure, we performed XRD analysis (Figure 2a). The high sharpness and intensity of the experimental peaks are remarkable indicators of the good crystalline nature of the material. Characteristic peaks of monoclinic Li_2_MnO_3_ (C2/m spatial group; COD entry # 96-154-4474) [39] can be distinguished. Thus, between 20${}^{\circ }$ and 25${}^{\circ }$, shoulder-like peaks related to the diffraction of the superlattice of Li_2_MnO_3_ are observed [40]. Moreover, the samples display the typical peak splitting between 63–67${}^{\circ }$, corresponding to the (-133) and (33-1) planes.

HRTEM was used for the local study of the microstructure of LMO. HRTEM images (Figure 2b) show the presence of agglomerated nanoparticles with granular morphology and particle sizes between 100 and 200 nm. Moreover, by analyzing the HRTEM image in Figure 2c, it is possible to identify the [001], [020] and [11-1] zone axes with interplanar spacings of 4.7 Å, 4.2 Å and 3.6 Å, respectively, which correspond to the layered structure Li_2_MnO_3_ [11,41].

### 3.2. Optimized Crystal Structure and Electronic Properties of the Li_2_MnO_3_

The Li_2_MnO_3_ crystal structure exhibits a monoclinic spatial ordering with a space group C2/m. The parameters of the conventional unit cell are a=4.933 Å, b=8.535 Å, c=5.026 Å and β=109.314° [42,43]. Herein, the Mn^+4^ ions are arranged in octahedral 4 g sites, while the lithium-ion, $Li^{+}$, are spread in 2c and 4h octahedral sites into the [$LiMn_{2}$] layer and [$Li^{+}$] layer stack [3]. Thus, we first calculated the structural parameters of the relaxed conventional unit cell of the Li_2_MnO_3_ by using the LDA method and compared them to the experimental values (Table 1). On the one hand, the results showed that the calculated lattice parameters a, b and c were underestimated by the LDA method compared to the experimental values by approximately 3%. On the contrary, the β angle (the angle between axes a and c) was slightly overestimated. Despite the fact that LDA has been shown to underestimate the lattice parameters for several types of materials severely, our optimized lattice parameters for the Li_2_MnO_3_ present good agreement with the experimental results.

Figure 3 shows the predicted projected density of states (pDOS) and band structure of the Li_2_MnO_3_ along with the high-symmetry points of the first Brillouin zone. The Fermi energy level ($E_{F}$) has been set to zero energy. The electronic DOS of the Figure 3 shows an overlap/hybridization between the 3d state of Mn and the 2p state of O, which is characteristic of 3d transition metal oxides [35]. The occupied states at the Fermi level are largely constituted of O 2p, while the Mn 3d orbitals contribute more to the conduction bands. The band gap between the lowest conduction and maximum valence levels are 1.65 eV and 1.99 eV for the spin-up and down configuration, respectively, thus assuming a semiconductor behavior. On the other hand, the electronic band structure of Li_2_MnO_3_ indicates that both valence band maximum and conduction band minimum are localized at the Γ-point.

### 3.3. Vibrational Properties of Li_2_MnO_3_

The computed phonon bands dispersion and phonon DOS for Li_2_MnO_3_ along the high-symmetry directions of the first Brillouin zone are shown in Figure 4. Importantly, the absence of imaginary frequency modes in the Brillouin zone demonstrates the structural dynamical stability of our model of Li_2_MnO_3_ [44,45]. According to the dispersion curves in Figure 4, the highest wavenumber is about 700 cm^−1^. More interestingly, the phonon band structure of Li_2_MnO_3_ does not contain a phonon gap [45]. The phonon DOS for Li_2_MnO_3_ is shown on the right-hand of Figure 4. The shaded green, red, and blue areas represent Li, O, and Mn atom phonon partial DOS, respectively. On the one hand, it is evident that the O atom contributes to the overall phonon distribution, dominating the zone with the highest intensities at higher frequencies. Additionally, the phonon distribution of the Mn atom extends to the whole spectrum but with lower intensity than the O atom. Interestingly, there is a hybridization between Mn and O vibrational states at higher frequencies. On the other hand, the phonon distribution of the Li atom mainly contributes to the middle part of the spectrum, ranging from 200 cm^−1^ to 500 cm^−1^, suggesting hybridization between Li and O vibrational states in this zone.

The unit cell of Li_2_MnO_3_ has 24 atoms. After the software analysis, there are three acoustic and sixty-nine optical modes. Li_2_MnO_3_ belongs to the C2/m space group (point group $C_{2h}$), and its irreducible representations at the Brillouin zone point are the following:(1)$$\Gamma_{acoustic}=A_{u}+2B_{u}$$
(2)$$\Gamma_{optic}=15A_{u}+12A_{g}+24B_{u}+16B_{g}.$$

Table 2 and Table 3 show the calculated Γ zone-centered optical phonon frequencies of Li_2_MnO_3_. In Li_2_MnO_3_, according to the $C_{2h}$ point group character table, the $A_{u}$ and $B_{u}$ modes are IR active, whereas the $A_{g}$ and $B_{g}$ modes are Raman active. The lowest and highest phonon frequencies are 173.38 and 668.64 cm^−1^, respectively, which is consistent with our previous results of the computed phonon dispersion.

The experimental and theoretical IR and Raman spectra are shown in Figure 5. The IR and Raman spectra calculated for Li_2_MnO_3_ were based on lattice dynamic simulations. As has been previously shown for other transition metal oxides, in our case, the number of experimental modes is much lower than that expected theoretically due to the three possible circumstances: (i) certain modes have very similar energy; (ii) stronger bands may overlap some weaker bands, and (iii) in spectra of a randomly oriented crystalline material, the relevant bands cannot be resolved properly [46,47].

Figure 5a shows the FT-IR spectra of synthesized Li_2_MnO_3_ powder. The sample exhibited strong absorption bands around 500–700 cm^−1^. Specifically, these bands appear at 514, 537 and 627 cm^−1^. The major contribution of these bands is related to the asymmetric stretching vibrations of octahedral MnO_6_ [21,25]. In this range, however, the stretching vibrations of tetrahedral LiO_4_ groups have also been observed at 650 cm^−1^ [19] and 350–550 cm^−1^ [23]. IR spectroscopy investigations of lithium inorganic compounds have demonstrated that the vibrations of MnO_6_ groups are tightly related to LiOn polyhedral vibrational modes at low and high frequencies (LiO_6_ and LiO_4_) [21]. The calculated IR spectra of Li_2_MnO_3_ are shown in Figure 5b. Interestingly, the theoretical spectrum shows similar characteristics to the experimental one, with some shifts in the frequency of the main vibrational modes. The results show two IR active modes with high intensity at higher frequencies (550–650 cm^−1^). Additionally, four intense IR active modes appear at the middle-to-low frequencies zone (250–510 cm^−1^). The absorption bands around 550–650 cm^−1^ and 250–510 region are attributed mainly to the MnO_6_ and LiOn functional groups, respectively, as can be seen in the atomic displacements for the IR vibrations modes of Li_2_MnO_3_ (Figure 6). Remarkably, these theoretical results are in agreement with the phonon study, which shows a strong hybridization between Li-O and Mn-O at low and high frequencies, respectively.

Figure 5c shows the typical Raman spectra of the synthesized Li_2_MnO_3_ powder. Eight major contributions are resolved with clarity. Strong bands at high frequencies, related to the stretching of Mn-O bond modes that arise from the MnO_6_ octahedra, are evidenced [48]. Additionally, weak bands in the low-frequency region related to Li-O bonds are observed [49]. The Raman spectra exhibited active Raman bands around 616, 574, 501, 442, 419, 374, 335, and 250 cm^−1^. Similar results were reported by other authors [49,50]. The calculated Raman spectrum of Li_2_MnO_3_ is shown in Figure 5d. Interestingly, the theoretical spectrum shows similar characteristics to the experimental one, showing minor differences in intensities. The results show eight Raman signals. Two main Raman active modes were evidenced at high and middle frequencies around 620 and 522 cm^−1^. These bands can be assigned to $A_{g}$ and $B_{g}$ modes due to the vibrations of the Mn atoms and the vibrations of Li atoms in the Mn layer, respectively [50,51]. Two representative atomic displacements for the Raman vibration modes of Li_2_MnO_3_ are shown in Figure 6. Interestingly, our phonon and Raman simulations are in agreement.

## 4. Conclusions

In the present study, we have systematically computed the phonon structure, infra-red, and Raman spectra of Li_2_MnO_3_ by first-principles calculations. We analyzed its crystal structure, optical phonon frequencies, IR, and Raman active mode and compared them to the experimental findings. By using the LDA exchange functional, we have found an underestimation of the lattice parameters by approximately 3% compared to the experimental one. Moreover, the electronic properties evidenced a semiconductor behavior with a strong hybridization between Mn 3d and O 2p states near the Fermi level. From the point of view of the vibrational properties, the phonon band structure of Li_2_MnO_3_ does not show a phonon gap. Additionally, the phonon distribution shows that Mn and O vibrational states dominate at higher frequencies, and Li and O atoms mainly contribute to the middle-to-low part of the spectrum. Simulated IR and Raman spectra agree with the experimental data. Thus, this work can be considered helpful for future studies dealing with the vibrational characterization of Li_2_MnO_3_ material.

## Figures and Tables

**Figure 1 materials-15-06237-f001:**
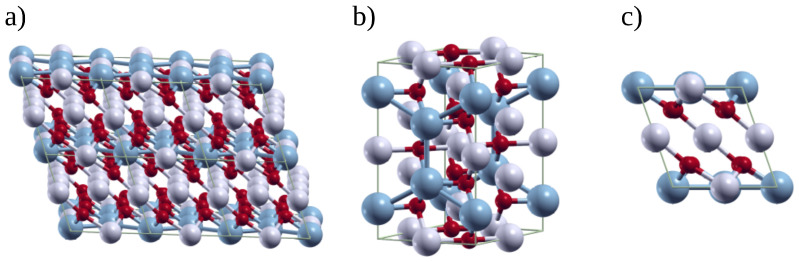
(**a**) A 2 × 1 × 2 super-cell showing the [Li^+^] and [$LiMn_{2}$] layers, (**b**) the conventional unit cell, and (**c**) the top view of the monoclinic Li_2_MnO_3_ crystal (Li: grey, Mn: blue and O: red).

**Figure 2 materials-15-06237-f002:**
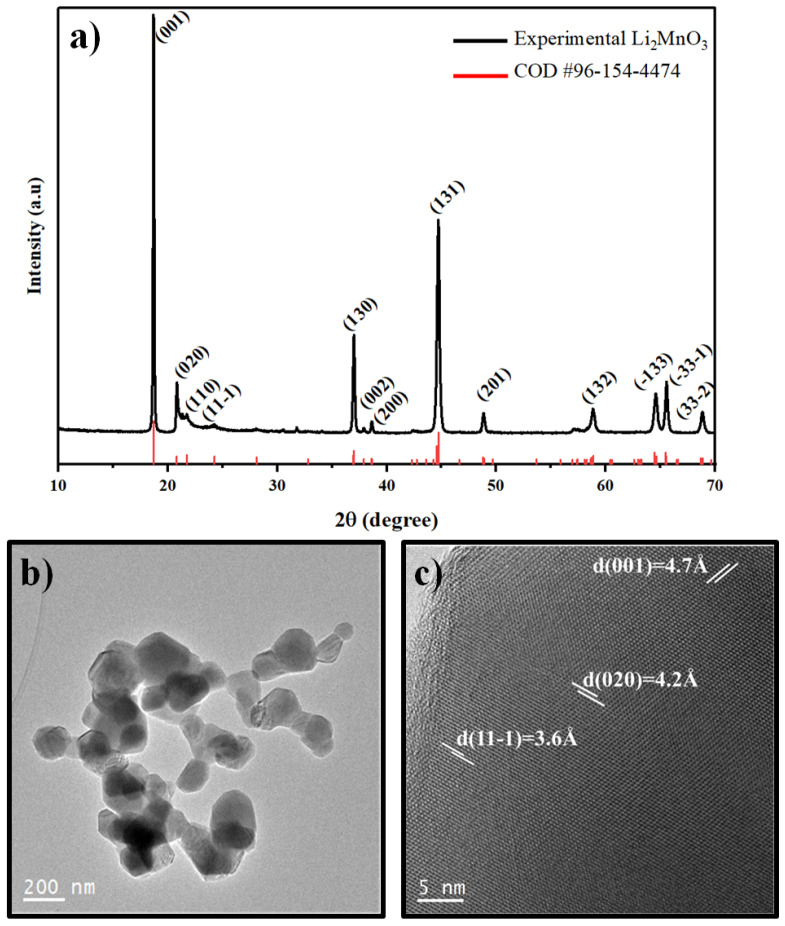
(**a**) X-ray diffraction pattern and (**b**,**c**) Transmission Electron Microscopy images of the Li_2_MnO_3_ powder material used in this study.

**Figure 3 materials-15-06237-f003:**
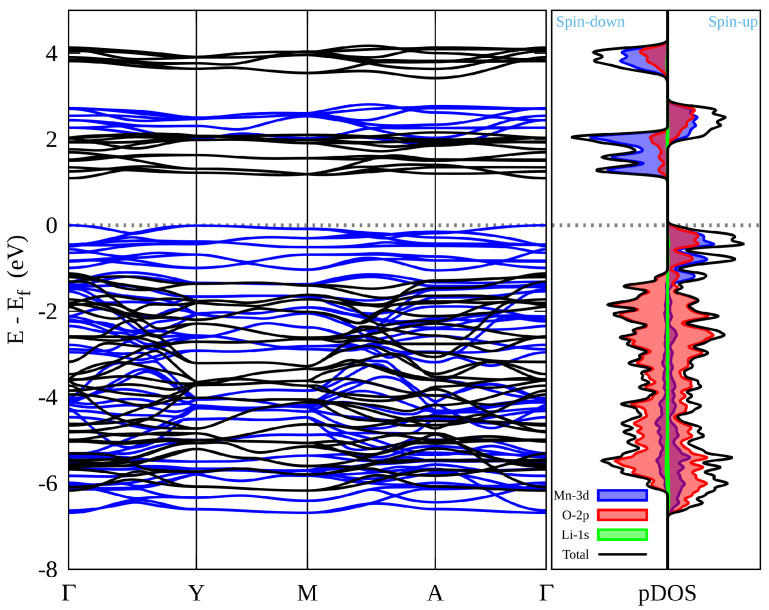
Electronic band structure and electronic density of states of Li_2_MnO_3_ calculated by LDA. In the electronic band structure, the blue and black bands correspond to the spin-up and down, respectively.

**Figure 4 materials-15-06237-f004:**
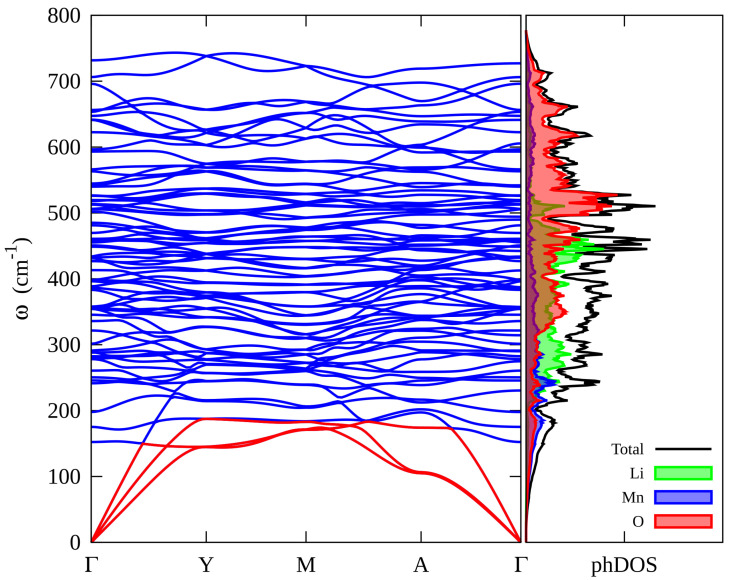
Phonon dispersion and phonon DOS for Li_2_MnO_3_ calculated by using the LDA method. In the phonon band structure, the acoustic bands are plotted in red.

**Figure 5 materials-15-06237-f005:**
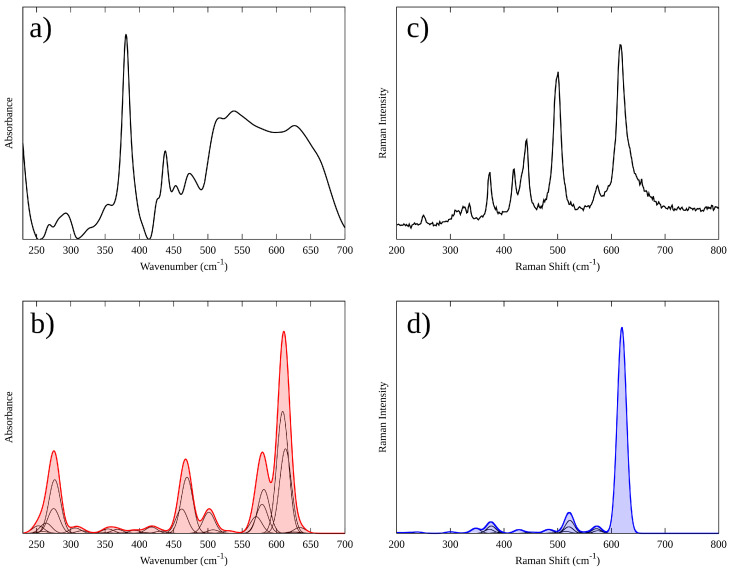
Experimental and theoretical IR (**a**,**b**) and Raman (**c**,**d**) spectra of Li_2_MnO_3_.

**Figure 6 materials-15-06237-f006:**
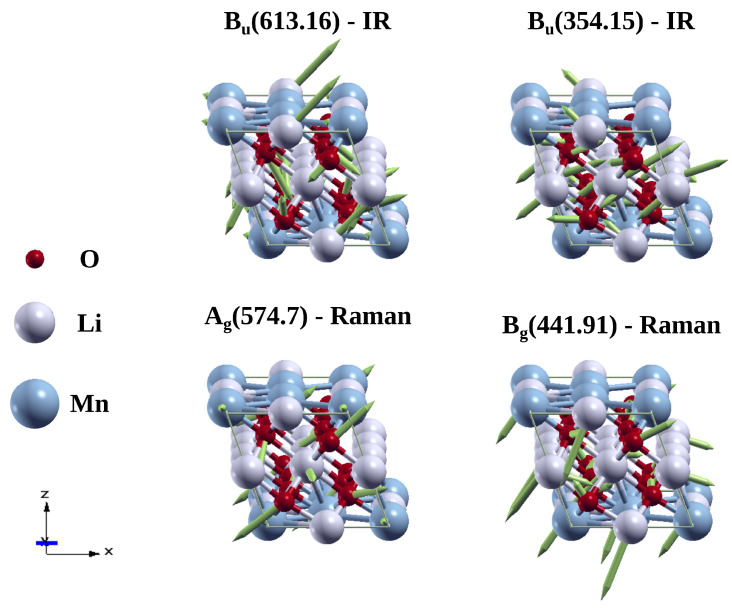
Atomic displacements for representatives IR and Raman vibrations modes of Li_2_MnO_3_ calculated by LDA.

**Table 1 materials-15-06237-t001:** Optimized structural parameters of the Li_2_MnO_3_ crystal determined by DFT calculations.

	a (Å)	b (Å)	c (Å)	β (°)	Volume (Å³)
LDA	4.805	8.308	4.854	109.628	182.494
Experimental	4.933	8.535	5.026	109.314	211.608

**Table 2 materials-15-06237-t002:** Experimental and calculated IR modes for the Li_2_MnO_3_ crystal determined by DFPT calculations.

Exp. IR freq.(cm^−1^)	Calc. IR freq.(cm^−1^)	Symmetry	Exp. IR freq.(cm^−1^)	Calc. IR freq.(cm^−1^)	Symmetry
	173.38	$B_{u}$		417.7	$B_{u}$
	181.89	$A_{u}$	425	431.79	$A_{u}$
	184.75	$B_{u}$		439.87	$B_{u}$
	208.99	$B_{u}$	437.8	461.78	$A_{u}$
	226.19	$B_{u}$	437.8	469.48	$B_{u}$
	252.46	$A_{u}$		489.38	$A_{u}$
	258.53	$B_{u}$		498.5	$B_{u}$
268	263.69	$B_{u}$		501.37	$A_{u}$
282	274.87	$A_{u}$	514	507.57	$B_{u}$
282	276.48	$B_{u}$		530.38	$B_{u}$
292	306.25	$B_{u}$		543.95	$B_{u}$
327	316.27	$A_{u}$	537	570.37	$A_{u}$
	334.01	$A_{u}$	537	579.04	$B_{u}$
352	350.02	$B_{u}$	537	581.66	$B_{u}$
352	354.15	$B_{u}$	627	609.12	$A_{u}$
	367.76	$B_{u}$	627	613.16	$B_{u}$
380	378.5	$B_{u}$	627	622.42	$A_{u}$
	393	$A_{u}$	627	628.49	$A_{u}$
	401.78	$A_{u}$	627	633.94	$B_{u}$
	407.29	$B_{u}$			

**Table 3 materials-15-06237-t003:** Experimental and calculated Raman modes for the Li_2_MnO_3_ crystal determined by DFPT calculations.

Exp. Raman freq.(cm^−1^)	Calc. Raman freq.(cm^−1^)	Symmetry	Exp. Raman freq.(cm^−1^)	Calc. Raman freq.(cm^−1^)	Symmetry
	201.07	$A_{g}$		482.56	$A_{g}$
	218.24	$A_{g}$	501	493.99	$B_{g}$
	224.08	$B_{g}$	501	514.9	$B_{g}$
250	238.84	$B_{g}$	501	520.66	$A_{g}$
309	300.07	$B_{g}$	501	522.63	$A_{g}$
324	322.81	$B_{g}$		549.63	$B_{g}$
335	347.34	$B_{g}$		552.83	$B_{g}$
374	372.87	$A_{g}$		565.07	$B_{g}$
374	376.47	$A_{g}$	574	572.2	$B_{g}$
	390.11	$A_{g}$	574	574.7	$A_{g}$
419	427.5	$A_{g}$	616	619.61	$A_{g}$
	435.11	$B_{g}$		623.26	$A_{g}$
442	441.91	$B_{g}$		668.64	$A_{g}$
	454.22	$A_{g}$			

## Data Availability

The data that support the findings of this study are available from the corresponding author upon reasonable request.

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
