# Peer review of "Phonon Structure, Infra-Red and Raman Spectra of Li2MnO3 by First-Principles Calculations"

_materials, 2022, doi:10.3390/ma15186237_

Round 1
Reviewer 1 Report
This paper systematically studied the electronic structure of Li2MnO3 by employing density functional theory (DFT) calculations. The unit cell lattice constants are relaxed at the LDA level. It is well-known that LDA underestimates the lattice constants by around 5%. So, the error in this work is within the acceptable range. The band structures and density of states are investigated to reveal the occupation of the bands near the Fermi level. The author has conducted a detailed study on the vibrational properties by employing DFPT implemented in QE. Both IR and Raman active modes are listed. The eigenvectors are clearly represented. This article could be a remarkable reference for the study of Li adsorption in Li2MnO3. This paper can be published with a few minor revisions to be done.
1. Line 86. The description of the k-point mesh is a bit misleading. Is it the k-point mesh for the Li2MnO3 conventional cell or the Li8Mn4O12 supercell?
2. Table 1. Since it is only a comparison between experimental and LDA relaxed structures, I suggest removing the header ‘XC Functional’. It is not a comparison between different exchange-correlation functionals.
3. Figure 3. PDOS revealed that CBM is dominated by the spin-up orbitals and VBM is dominated by the spin-down orbitals. So, the direct band gap cannot be confirmed as the transition from alpha orbital to beta orbital is spin forbidden. I suggest splitting the band structures by presenting the spin-up and -down eigenvalues separately.
4. What is the converged magnetic moment for Mn(IV) when considering the ferromagnetic configuration for Li2MnO3? Is ferromagnetic the spin ground state for L2MnO3?
5. The analysis of phonon dispersion is excellent. The IR active and Raman active modes have been listed in Table 2 and 3. Since there are too many optical branches, the acoustic branch dispersion is somehow not clear. Is it possible to draw the acoustic branches in another color?
6. The material has significant applications for Li batteries. How is the future to be conducted? The phonon dispersion can also be used to derive the thermal-electric properties. The Li transportation could also be important. As this article has done some preliminary study, will the ad/desorption of Li be studied in the future?
Author Response
Dear reviewer,
You will find attached to this message the point-by-point response to yout comments .

Reviewer 2 Report
The manuscript reports a first-principles calculation about the Phonon structure, infra-red and Raman spectra of Li2MnO3. The calculated results are supported by experiments to a certain extent. The following comments should be considered before further publication:
1. As we know, Generalized Gradient Approximation (GGA) functional is more popular, what is the reason for the authors to use Local Density Approximation (LDA) in this study, and is there any literature support?
2. Did the authors consider the d-orbital electron strong interaction of Mn, i.e. the U value. Due to the difference (3%) between the calculated lattice parameters and the experimental ones, the rationality of the calculated parameters needs to be further considered.
3. It is recommended to provide a standard PDF card for comparison in the XRD spectrum.
4. Some formatting issues require attention, such as subscripting of material formulas.
Author Response
Dear reviewer,
You will find attached to this message the point-by-point response to tyour comments.

Round 2
Reviewer 1 Report
The author has finished all necessary revisions. The spin-resolved band structure is presented decently. The paper is ready to be published.
Author Response
We gratefully appreciate your valuable comment
Reviewer 2 Report
I am satisfied with this revision and recommend the acceptance for publishing in Materials.
Author Response

(The authors gave the same response as above.)
